Prognostic implications of cell division cycle protein 45 expression in hepatocellular carcinoma

Yang Chen 1 2
Xie shufang 2 3
Wu Yi 4
Ru Guoqing 5
He Xianglei 5
Pan Hong-yin 6
Wang Shibing wangshibing@hmc.edu.cn 2 7
Tong Xiangmin tongxiangmin@163.com 2 7
1 Department of Clinical Medicine, Qingdao University , Qingdao , China
2 Molecular Diagnosis Laboratory, Zhejiang Provincial People’s Hospital , Hangzhou , China
3 The Second Clinical Medical College, Zhejiang Chinese Medical University , Hangzhou , China
4 Phase I Clinical Research Center, Zhejiang Provincial People’s Hospital , Hangzhou , China
5 Department of Pathology, Zhejiang Provincial People’s Hospital , Hangzhou , China
6 Department of Infectious Diseases, Zhejiang Provincial People’s Hospital , Hangzhou , China
7 The Key Laboratory of Tumor Molecular Diagnosis and Individualized Medicine of Zhejiang Province, Zhejiang Provincial People’s Hospital , Hangzhou , China
Albertini Maria Cristina
Electronic publication date: 2021 Feb 12
Publication date: 2021
Volume: 9
Electronic Location ID: e10824
Received 2020 Sep 9; Accepted 2021 Jan 3
Copyright: ©2021 Yang et al.
Copyright year: 2021
Copyright holder: Yang et al.
License: This is an open access article distributed under the terms of the Creative Commons Attribution License, which permits unrestricted use, distribution, reproduction and adaptation in any medium and for any purpose provided that it is properly attributed. For attribution, the original author(s), title, publication source (PeerJ) and either DOI or URL of the article must be cited.
License URL: https://creativecommons.org/licenses/by/4.0/

Keywords: Cell division cycle protein 45, Hepatocellular carcinoma, Biomarker, Bioinformatics

Funding: Zhejiang Medical Technology Plan Project WKJ-ZJ-1709 Zhejiang Provincial Natural Science Foundation of China LY19H160037 LGF18H160025 LY17H160062 2017C33116 This study was supported by the Zhejiang Medical Technology Plan Project (No. WKJ-ZJ-1709, 2020KY052) and the Zhejiang Provincial Natural Science Foundation of China (No. LY19H160037, LGF18H160025, LY17H160062, 2017C33116). The funders had no role in study design, data collection and analysis, decision to publish, or preparation of the manuscript.

==============================
Background

The overall prognosis of hepatocellular carcinoma (HCC) is poor and novel prognostic biomarkers might better monitor the progression of HCC. Cell division cycle protein 45 (CDC45) plays a key role in DNA replication and considered to be involved in tumorigenesis. This study investigated CDC45 expression in tumour tissues and defined its prognostic value in HCC patients.

Methods

We used immunohistochemistry (IHC) staining to examine the expression of CDC45 in tumour tissue specimens and compare them with adjacent normal tissue specimens using a constructed tissue microarray (TMA) and analyzed how clinical features are related to HCC prognosis. Functional enrichment analyses were used to describe significantly involved hallmark pathways of differentially expressed genes (DEGs, which were screened out according to the high or low expression of CDC45 in tumour tissues).

Results

Our findings showed that the proteome expression of CDC45 was evidently downregulated in HCC tissues compared with matched normal tissues (P < 0.0001). Although we did not find any differences in terms of vascular invasion, metastasis, lymphatic infiltration, or Edmondson grade between patients with high and low CDC45 expression, low CDC45 expression was significantly correlated with microvascular invasion (P = 0.046). Multivariate analysis indicated that CDC45 expression (P = 0.035) was an independent prognostic factor for the overall survival (OS) rate of HCC patients. Patients with CDC45 expression was positively correlated with OS rates among HCC patients (P < 0.05). Functional annotations indicated that CDC45 is involved in the most significant pathways, including the cell cycle, DNA replication, chemical carcinogenesis and drug metabolism–cytochrome P450 pathways.

Discussion

Our findings showed that low proteomic level of CDC45 was associated with a poor prognosis in HCC patients, indicating that CDC45 might be a novel prognostic marker.

Introduction

Hepatocellular carcinoma (HCC) is the most common type of primary liver cancer and ranks as the fourth leading cause of cancer mortality all over the world (Bray et al., 2018). In the last few years, its incidence has sharply increased (Allemani et al., 2018). Although some curative therapies, including hepatic resection, liver transplantation and the application of targeted drugs, can prolong survival for HCC, the 5-year OS for HCC patients is quite disappointing (Kardashian et al., 2020; Heimbach et al., 2018). In addition, the heterogeneity of liver cancer hinders choosing the optimal treatment for patients (Dong et al., 2019; Losic et al., 2020). The biomarkers that can primely predict the prognosis may guide treatment in HCC patients and improve HCC clinical outcomes. Nevertheless, the specificity and sensitivity of current biomarkers still need to be improved (Nault & Villanueva, 2020). Therefore, it is particularly critical to identify effective biomarkers for the diagnosis and prognosis of patients with liver cancer, which might provide the opportunity to use targeted drugs and immune modulators earlier in the disease course.

CDC45 is an integral component of the CDC45-MCM2-7-GINS (CMG) helicase complex, which plays a crucial role in DNA replication, especially in the initiation stagey (Simon et al., 2016; Broderick & Nasheuer, 2009). Even it is still unclear whether CDC45 plays any role in the repair pathways in mammalian cells, CDC45 plays an important biological role in maintaining the genomic stability (Aze et al., 2013; Errico & Costanzo, 2012). Furthermore, it has been indicated that CDC45 is important for the response to replication stress by interacting with ssDNA (Bruck & Kaplan, 2013) and recruiting Rad53 (Can et al., 2019). Recently, CDC45 expression was identified to associate with tumorigenesis and was useful for tumour prognosis. Previous studies have shown high CDC45 expression in human cancer-derived cell lines, including breast carcinoma, cervix carcinoma and acute lymphoblastic leukaemia (Pollok et al., 2007). Moreover, it was demonstrated that the upregulation of CDC45 in papillary thyroid cancer is correlated with a more advanced tumour stage (Sun et al., 2017). In non-small cell lung cancer, CDC45 can promote the growth of tumour cells and was defined as an oncogene (Huang et al., 2019). However, rare studies have investigated the relationship between CDC45 expression and predictive value in HCC.

In the present study, we examined the protein expression level of CDC45 in HCC by comparing tumours with neighbouring normal tissues in tissue microarrays (TMAs). Then, a larger sample size was used to investigate the association of CDC45 expression and clinical characteristics of patients with HCC. Additionally, we analysed the potential biological interaction network and prognostic value of CDC45. We hypothesized that the possible antioncogenic activity of CDC45 may influence the HCC patients’ prognosis. Our results may uncover a feasible therapeutic target and provide insights into the molecular mechanisms of CDC45.

Materials and Methods

Patient characteristics and construction of the tissue microarray

TMAs from 321 HCC patients who took curative hepatectomy for primary HCC in Zhejiang Provincial People’s Hospital between 2008 and 2015 were used for validation. The ethics committee of Zhejiang Provincial People’s Hospital granted Ethical approval to carry out the study within its facilities (Ethical Application Ref:2019KY232). All patients did not receive radiotherapy or chemotherapy before surgery. The whole group of patients were informed and signed informed consent forms. Total survival was calculated from the date of surgery to the end of follow-up (December 2016) or the time of death. In total, 321 HCC patients in our study presented with a mean age of 56.9 ± 11.4 years (range, 25.0–90.0 years), with 60 (18.7%) females and 261 (81.3%) males (Table 1). At the time of the primary diagnosis, 52.1% of the patients presented with a tumour diameter less than 5 cm, 47.9% with a tumour diameter greater than or equal to 5 cm. A total of 63.2% had an Edmondson grade of I/II, and 36.8% had grade III. In addition, the numbers of patients with or without metastasis were 288 (91.4%) and 27 (8.6%), respectively. The median OS was 23.0 months (range, 1.0–74.0). Other clinical characteristics are displayed in Table 1.

Table 1 Association between the expression of CDC45 expression and clinical characteristics of HCC patients.

Clinicopathological variables	n (%)	CDC45 expression	P-value	
		Low ()	High ()		
Sex				0.213	
Male	261 (81)	129	132		
Female	60 (19)	35	25		
Age (years)				0.544	
<55	124 (39)	66	58		
≥55	197 (61)	98	99		
Size (cm)				0.034	
<5	163 (52.1)	75	88		
≥5	150 (47.9)	87	63		
Tumour number				0.797	
Single	264 (82)	134	130		
Multiple	57 (18)	30	27		
Edmondson grade				0.299	
I–II	199 (63.2)	96	103		
III	116 (36.8)	63	53		
Metastasis				0.306	
M0	288 (91)	147	141		
M1	27 (9)	11	16		
Vessel invasion				0.169	
Absence	120 (50)	62	58		
Presence	119 (50)	72	47		
Microvascular invasion				0.046	
Negative	133 (57.6)	68	65		
Positive	98 (42.4)	63	35		
HBs antigen				0.57	
Negative	59 (19)	28	31		
Positive	256 (81)	132	124		
Cirrhosis				0.971	
Negative	106 (33)	54	52		
Positive	215 (67)	110	105		
Note.

Bold signifies p value<0.05.

Immunohistochemical staining

IHC was performed using standard techniques. Briefly, 5-µm paraffin-embedded TMA sections were dewaxed using xylene (Sinopharm, China)and rehydrated in graded alcohols (Sinopharm, China). Next, blocked endogenous peroxidase with 3% hydrogen peroxide (sinopharm, China). Antigen retrieval was accomplished via adding 10 mM citrate buffer (pH 6.0) (Sinopharm, China) to the TMA sections and putting them into a high-pressure cooker. Then, the TMA sections were incubated with 1% bovine serum albumin (BSA; Sigma, German) for 20 min to reduce nonspecific protein binding. The recombinant rabbit monoclonal CDC45 antibody (1:50; HuaBio, Hangzhou, China) were next used to treat the TMA sections at room temperature for 1 h. They were then rinsed with phosphate-buffered saline (PBS; HuaBio, Hangzhou, China) and incubated with biotinylated secondary antibody (MXB, Fuzhou, China) for 30 min at room temperature. Subsequently, the TMA sections were stained with DAB chromogen (Gene Tech, Shanghai, China), counterstained with Mayer’s haematoxylin (HuaBio, Hangzhou, China), dehydrated with gradient alcohol and xylene, and mounted on slides. Finally, the TMA sections were viewed under a Nikon light microscope.

Cell culture

Human normal liver cell line LO2 and hepatocellular carcinoma cell lines (HCC-LM3, MHCC-97H, Huh7 and Hep3b) were purchased from the ATCC and cultured in Dulbecco’s modified Eagle’s medium (DMEM, Gibco, USA) containing 10% fetal bovine serum (Gibco, USA) and 1% penicillin/streptomycin (Beyotime, China) in a humid atmosphere with 5% CO 2 at 37 °C. All cells were cultured at a density of 4 ×105 cells/ml during the experiments. Approximately 8 ×105 cells were plated on a 6-well plastic cells (Nest, Jiangsu, China) and then collected for western blot.

Western blot and antibodies

Cells were washed with PBS (Beyotime, China) twice. Next, RIPA buffer were used to lyse the cells. Thereafter, BCA Protein Quantitation Assay (Beyotime, China) was used to detected the protein concentration of the cells. Proteins were resolved using 10% SDS-PAGE and then transferred onto a nitrocellulose membrane. After block with 5% non-fat milk for 1 h at room temperature, the membranes were then incubated with the primary antibodies at 4 °C overnight. After that, TBST was used to wash the membranes three times and subsequently incubated with secondary antibodies (anti-rabbit IgG or anti-mouse IgG) for 1 h at room temperature. Next, membranes were washed with TBST three times. The targeted proteins were identified using the ECL (Thermo Fisher, Waltham, MA, USA). CDC45 and ACTIN antibodies used in study were purchased from HuaBio (ET1701-71, China) and Abcam (ab8227) respectively.

Assessment of CDC45 expression

The Densito Quant software in Quant Center was applied to automatically identify and set the areas on the tissue sections as follows: dark brown as strong positive, tan as moderate positive, blue (nucleus) as negative, and light yellow as weak positive. Then, each tissue point was identified and analysed to find the areas of strong positive, medium positive, weak positive and negative (unit: pixel); then, the percentage of positive staining and the histochemistry score (H-score) were calculated. The median H-score (median = 7.44) was selected as the cut-off value to classify the level of CDC45 expression. H-scores <7.44 were considered as low CDC45 expression, and H-scores >= 7.44 were used to define tumours with high CDC45 expression.

Bioinformatics analysis of GSE76427

We used the Gene Expression Omnibus (GEO, http://www.ncbi.nlm.nih.gov/geo/) database, an international public platform including high-throughput gene expression and other functional genomics datasets for the research community (Barrett et al., 2013). We selected the GSE76427 (Grinchuk et al., 2018) dataset and downloaded the original CEL files as well as the platform files to explore CDC45 expression and its potential mechanisms by means of R software. Specifically, according to the median value of the CDC45 gene among HCC tissues, samples were divided into two groups including high or low expression. Then, the adjusted p-value <0.05 and —log2FC—>1.5 were used as cut-off criteria to screen differentially expressed genes (DEGs) between the two groups of samples. Subsequently, we conducted Gene Ontology (GO) and Kyoto Encyclopedia of Genes and Genomes (KEGG) pathway enrichment analyses with DEGs. In addition, the protein–protein interaction (PPI) network of DEGs was constructed with a online search tool for the retrieval of interacting genes (STRING; http://string-db.org), and to analyse the functional interactions between proteins (an interaction with a combined score >0.9 was considered statistically significant), which were then visualized using Cytoscape. Moreover, we verified important modules via using the molecular complex detection (MCODE) app plugin of Cytoscape.

Statistical analysis

All data were analysed using SPSS 22.0 software (SPSS Inc., Chicago, IL, USA). The univariate and multivariate hazards were determined by Cox proportional hazards model. Differences between HCC and noncancerous tissues were examined by paired t-test. The correlation results between the clinicopathological parameters and the CDC45 expression were investigated by the chi-square analysis. The comparation of CDC45 expression among different cell lines was examined by t-test. We used Kaplan–Meier analysis to estimate survival, and any differences in survival were evaluated with a stratified log-rank test. Generally, P < 0.05 was considered statistically significant.

Results

Our research consisted of three stages. Firstly, we assessed protein expression of CDC45 in our TMA. Then we investigated the association of CDC45 expression with clinical characteristics and its prognostic value in HCC. In the third stage, we screened the significantly involved DEGs of CDC45 and performed corresponding functional annotations using DEGs.

Validation of CDC45 expression in HCC and adjacent normal tissues

We use TMA to detected the protein expression levels. IHC staining was used to evaluate the CDC45 protein expression levels in HCC tumour samples and matched normal tissues from 56 HCC patients (Fig. 1A). CDC45 staining was mainly distributed in the cytoplasm. Compared to side normal tissues, the protein expression of CDC45 was significantly decreased in HCC tissues (P < 0.0001, Fig. 1B).

Figure 1 CDC45 protein expression and prognostic implications in the TMA of HCC.

(A) IHC staining indicated significantly downregulated CDC45 expression in HCC tissues comparing to the adjacent normal liver tissues in the TMA. (B) The differential CDC45 protein expression in 56 paired tumour and non-tumorous liver tissues (p < 0.0001). (C) Kaplan–Meier survival analysis indicated that low CDC45 expression was indicated a shorter OS (p = 0.019).

Association of CDC45 expression with clinical characteristics

We then explored the correlation between CDC45 expression levels and the clinicopathological parameters of HCC patients. Among the clinicopathological features, CDC45 expression was positively related with tumour size (P = 0.034) and microvascular invasion (P = 0.046). However, there was no significant correlation between CDC45 expression and other clinical parameters (gender, age, tumour number, hepatitis B surface antigen (HBS), Edmondson grade, vascular invasion, cirrhosis and metastasis, P > 0.05; Table 1). These findings suggested that low CDC45 expression is associated with worse overall disease condition.

Clinical values of CDC45 expression in the prognosis of HCC

The Kaplan–Meier survival curve revealed that CDC45 expression was evidently associated with OS in HCC patients (P = 0.019; Fig. 1C). In all HCC patients, the OS was shorter in patients with low CDC45 expression than those who with high CDC45 expression. Additionally, Cox regression was used to analyse the prognostic factors of HCC. Our univariate Cox regression revealed that tumour size (P = 0.007), metastasis (P < 0.001), microvascular invasion (P = 0.022), Edmondson grade (P < 0.001) and vessel invasion (P = 0.011) were independent prognostic factors in patients with HCC. A multivariate Cox regression analysis revealed that distant metastases (P < 0.001) and Edmondson grade (P = 0.013) were independent prognostic factors for patients with HCC (Table 2). Consistent with the results of the Kaplan–Meier analysis, both the univariate (P = 0.022) and multivariate (P = 0.035) Cox regression analysis indicated that CDC45 expression was closely associated with the prognosis of HCC.

Table 2 Univariate and multivariate Cox regression survival analyses.

Parameters	Univariate analyses	Multivariate analyses	
	Hazard ratio (95% CI)	P-value	Hazard ratio (95% CI)	P-value	
CDC45 expression	0.558 (0.338–0.920)	0.022	0.509 (0.271–0.957)	0.035	
Size	1.965 (1.201–3.213)	0.007	1.157 (0.614–2.177)	0.652	
Metastasis	5.075 (2.673–9.636)	0.000	6.533 (2.916–14.635)	0.000	
Microvascular invasion	1.879 (1.097–3.219)	0.022	1.015 (0.374–2.753)	0.977	
Vessel invasion	2.035 (1.174–3.525)	0.011	1.636 (0.602–4.440)	0.334	
Edmondson grade	2.696 (1.655–4.392)	0.000	2.082 (1.166–3.715)	0.013	
Note.

Bold signifies p value<0.05.

Figure 2 Enrichment analysis and PPI network construction among DEGs.

(A) Distributions of up- and downregulated DEGs are shown in the volcano plot. The blue colour represents the downregulated genes, while the yellow colour represents the downregulated genes. (B) The heat map shows the top 20 significant DEGs with positive and negative correlations with CDC45. (C) The 30 most enriched GO terms of the DEGs were obtained in HCC patients. (D) Top 30 enriched KEGG pathways of the DEGs are shown in the bar plot.

Identification of DEGs and functional annotations as well as predicted signalling pathways

To further study the potential biological function of CDC45 in HCC, the genes from the GSE76427 dataset were divided into two groups based on the median value of CDC45 expression to screen the DEGs between the low and high CDC45 expression groups. The screening criteria are discussed in the Materials and Methods section. As shown in Fig. 2A, there were 233 upregulated and 167 downregulated DEGs. Specifically, the top 20 significant DEGs with positive and negative correlations were shown in Fig. 2B with a heat map. Subsequently, 400 involved DEGs were subjected to conduct GO annotation and KEGG pathway analyses. And top 30 GO terms with the highest gene enrichment are shown in Fig. 2C. Those related genes were significantly involved in eukaryote division included organelle fission, nuclear division, chromosome segregation, mitotic nuclear division, and were markedly involved in the regulation of the cell cycle phase transition and DNA replication. Moreover, the KEGG analysis showed that most of the involved significant pathways contained the cell cycle, DNA replication, drug metabolism and metabolism of xenobiotics by cytochrome P450, chemical carcinogenesis, meiosis and fatty acid degradation signalling (Fig. 2D). The specified functional annotations, KEGG analysis information and percentage of each term are shown in additional file 1.

To further explore the interplay among the DEGs, we used the STRING online database and Cytoscape software to construct a PPI network. As illustrated in Fig. 3A, the network includes 400 nodes and 1,622 edges. The nodes with positive correlation with CDC45 expression are shown as yellow, nodes with negative correlation are shown as blue, and CDC45 itself is purple. PPI network was then carried out clustering analysis by Cytotype MCODE, the top three significant modules were selected according to the degree of importance. Module 1 contains 43 nodes and 683 edges (Fig. 3B), which means that CDC45 interacts is closely related those 43 proteins. module 2 contains 13 nodes and 39 edges (Fig. 3C); and module 3 contains 19 nodes and 58 edges (Fig. 3D). Submodule analysis information is shown in additional file 2.

Figure 3 PPI network construction of DEGs and module analysis.

(A) Yellow nodes among the PPI network represent upregulated genes and blue nodes for downregulated genes, and purple nodes for CDC45. (B) Module 1 of the PPI network. (C) Module 2 of the PPI network. (D) Module 3 of the PPI network.

Discussion

CDC45 is a key component of the CMG complex and necessary for the initiation and elongation of chromosomal DNA replication (Simon et al., 2016; Saha et al., 1998). In recent years, several reports have shown that its expression is related to the occurrence and development of tumours. It has been reported that CDC45 overexpression was observed in lung cancer and was considered a novel tumour-associated antigen (TAA) that might be a useful target for lung cancer immunotherapy (Tomita et al., 2011). Furthermore, Hu et al. (2019) verified that the upregulated CDC45 expression in colorectal cancer patients was associated with poor prognosis. A previous study also revealed that the CDC45 protein level is consistently higher in various kinds of human tumour cells than in normal cells so that was identified as a proliferation-associated antigen (Pollok et al., 2007).

However, in our study, we found that CDC45 expression at the protein level was downregulated in HCC by detecting the expression of CDC45 in tumour tissues and matched adjacent normal liver tissues from 56 HCC patients. Although there are fewer samples in this study, this pairing analysis of tumour and adjacent normal liver tissues seems to be more convincing. Similarly, WB results showed that CDC45 expression was lower in HCC-LM3 cells, MHCC-97H and Huh7 cells than normal liver LO2 cells (FigureS1). Moreover, our results demonstrated that high protein expression of CDC45 negatively correlated with tumour size and metastasis, which revealed that decreased expression of CDC45 may promote cancer progression. Importantly, the Kaplan–Meier survival analysis indicated that HCC patients with a low expression of CDC45 have an inferior prognosis than patients with a high expression of CDC45. To further analyse the possible causes, we screened for DEGs related to CDC45 expression from the GEO database and conducted GO and KEGG analyses. Our results showed that the genes mostly enriched in the GO terms included organelle fission, nuclear division, chromosome segregation, mitotic nuclear division, regulation of cell cycle phase transition, regulation of mitotic cell cycle phase transition, nuclear chromosome segregation, sister chromatid segregation, DNA replication, and cell cycle G1/S phase transition. Furthermore, the KEGG analysis showed that most of the involved significant pathways included the cell cycle, DNA replication, drug metabolism–cytochrome P450, metabolism of xenobiotics by cytochrome P450, and chemical carcinogenesis. Therefore, we speculated that the correlation between CDC45 and HCC might be related to these functions.

Actually, it is common for the same gene to play different roles in various tumours. Typically, Sprouty2, which inhibits the growth and spread of tumour in breast cancer, prostate cancer and liver cancer, may contributes to the promotion of colorectal cancer metastasis (Zhang et al., 2016). Although it has been reported that CDC45 is a proliferative marker in some cancers, there are also reports stressed that the high expression of CDC45 may lead to severe replication pressure, subsequently occurring S-phase arrest, and eventually lead to cellular apoptotic cell death, inhibiting the degradation of CDC45 might be a promising way to combat cancer (Köhler et al., 2016; Zhang et al., 2016). In addition to the sustained proliferation, the escape from apoptosis, genomic instability, and DNA replication stress are also considered hallmarks of human cancers and are closely related to tumorigenesis and progression (Nayak, Calvo & Cantor, 2015). Therefore, high expression of CDC45 induced DNA stress of cancer cells, leading to cell death might explain the poor prognosis of low CDC45 expression in HCC patients.

This study has several limitations. First, we only conduct a single center study in our hospital, number of patients may bring some limitations, Secondly, even though a series of functional annotations and enrichment analyses were performed, the underlying mechanisms of signalling pathways in HCC remain ambiguous. Although the research has some limitations, we are committed to provide some basis for the biological function of CDC45 in liver cancer. In the future, we may expand the sample size and even conduct multi center research for better verify. In addition, to explore the detailed mechanism between CDC45 and carcinogenesis as well as reveal the mechanism of CDC45 in other cancers.

Conclusions

In conclusion, our research revealed the crucial role of CDC45 in patients with HCC. Decreased CDC45 protein expression correlated with the progression and poor prognosis in HCC, indicating that CDC45 is a valuable promising prognostic biomarker and might be a potential treatment target in HCC. Further studies are required to analyse its specific biological function in HCC.

Supplemental Information

File S1 KEGG analysis information and percentage of each term

Click here for additional data file.

File S2 GO analysis information and percentage of each term

Click here for additional data file.

File S3 PPI and submodule analysis information

Click here for additional data file.

File S4 Raw tissue microarray data

Click here for additional data file.

File S5 Code for Figs. 2A and 2B

Click here for additional data file.

Supplemental Information 6 Code for Figs. 2C and 2D

Click here for additional data file.

Supplemental Information 7 Original Fig. 3 data

Click here for additional data file.

File S6 Original CDC45 expression data in tumor and adjacent normal tissue

Click here for additional data file.

File S7 Original lasso regression data

Click here for additional data file.

File S8 Original cell experiment data

Click here for additional data file.

File S9 Survival analysis results of GSE 76427

Click here for additional data file.

We would like to thank American Journal Experts for editing the English version of this manuscript.

Additional Information and Declarations

Competing Interests

Author Contributions

Human Ethics

Data Availability

The authors declare there are no competing interests.

Chen Yang performed the experiments, analyzed the data, prepared figures and/or tables, authored or reviewed drafts of the paper, and approved the final draft.

shufang Xie, Yi Wu, Guoqing Ru, Xianglei He and Hong-yin Pan performed the experiments, analyzed the data, prepared figures and/or tables, and approved the final draft.

Shibing Wang and Xiangmin Tong conceived and designed the experiments, analyzed the data, authored or reviewed drafts of the paper, and approved the final draft.

The following information was supplied relating to ethical approvals (i.e., approving body and any reference numbers):

The ethics committee of Zhejiang Provincial People’s Hospital granted Ethical approval to carry out the study within its facilities (Ethical Application Ref:2019KY232).

The following information was supplied regarding data availability:

Raw data and code are available in the Supplemental Files.

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
