# Peer review of "Prognostic implications of cell division cycle protein 45 expression in hepatocellular carcinoma"

_PeerJ, doi:10.7717/peerj.10824_

## Round 0.1 · original submission · Major Revisions

Major concerns need to be addressed.

Reviewer 1 ·

Basic reporting

The authors conclude that low expression of CDC45 has a poor prognosis for HCC, but the mainstream paper is that high expression is associated with poor prognosis in most cancers. If HCC concludes that low expression of CDC45 has a poor prognosis, further confirmation by WB or PCR is required.

Experimental design

.

Validity of the findings

.

Additional comments

Data should be collected more carefully if it claims the opposite of what is pointed out in most cancers.

Reviewer 2 ·

Basic reporting

English language needs attention throughout the manuscript. There are a significant number of grammatical and logical errors so I suggest that the article is subject to English language editing. A few examples are list as below:

Line 27: Reword the sentence to ‘The overall prognosis of HCC is poor’. Also, exploring prognostic biomarkers is to monitor the progression/prognosis of HCC, rather than ameliorating the disease

Line 29: Delete ‘gradually’

Line 34: Revise the sentence to ‘analyzed how clinical features are related to HCC prognosis

Line 36: to describe the DEGs between HCC patients and normal control, and the related pathways.

Line 43: Reword the sentence to ‘CDC45 expression was positively correlated with OS rates among HCC patients’

Line 59: Prognostic biomarkers can only provide information/a prediction of disease progression, not treatment

Experimental design

Line 99-110: Please provide manufacturer information of all the reagents, solvents, and equipment

Line 141: How did you pair the samples? What statistical method did you use to compare the frequency of HCC stages?

Validity of the findings

Line 73: CDC45 was associated with poor prognosis in non-small cell lung cancer. However, you reported a positive association between CDC45 and HCC prognosis. Please discuss such discrepancy in the discussion section

Line 153: CDC45 will be exported out of the cell nucleus after DNA replication, as a method of preventing DNA re-replication. So the ratio of nuclear/cytoplasmic CDC45 is important. Please provide such information.

Figure 1B: The figure showed that the CDC45 score between HCC and normal control was not significantly different (the upper 25% SD bar of the ‘tumor group’ was higher than the 75% SD bar of the ‘control group’). However, the authors are claiming that the P-value was less than 0.0001. Please provide the original data.

Line 171-173: Please use another model to adjust for potential confounders (such as age, HCC stage, tumor size, etc.) in the multivariate Cox regression

Additional comments

The authors investigated the role of CDC45 in HCC prognosis and provided evidence that a lower level of CDC45 was associated with poor HCC prognosis. However, the authors need to improve scientific writing and provide extra information to validify their findings.

Reviewer 3 ·

Basic reporting

• As a reviewer I could find the translated medical ethics approval document
• I was not able to find raw deidentified data for Table1.

Experimental design

• All the 321 patients were not considered for Table1. Here Authors have to detail why all the 321 patients were not been included for each of the clinical pathological variables (ex: Tumor size etc). If they do not have data on some of them then they might have to include another section called “NA” in each of the variables.

• Similarly, inclusion and exclusion criteria were not described for survival cohort. As a reviewer I see that entire cohort is about 321 patients and in survival graph from figure 1C, i could only see 190 patients. Authors have to describe which set of patients were considered for this figure by describing inclusion & exclusion criteria.

• Survival graph description in manuscript should not be Fig 3c but it should be Fig 1c (line 166).

Validity of the findings

• Authors have described in study limitation that they could not do expression levels of CDC45 in transcriptome due to lack of samples but they haven’t explained why they have not considered doing a survival plot of CDC45 expression utilizing publicly available dataset such as GSE 76427. It would serve as validation of survival plot using protein expression and make the study robust overall.

Additional comments

I appreciate authors for putting great deal of effort into this study that details of CDC45 in HCC. This manuscript details a combination of ex vivo studies using tumor, normal tissues and bioinformatics analysis of publicly available dataset to discuss novel findings of CDC45 linked to poor prognosis of HCC.

·

Basic reporting

a) This paper is well-organized with good background material. I found no significant grammar issues.

b) Although there are no studies exactly investigating CDC45 as a prognostic marker in HCC, can authors add a couple of studies (PMID: 19136905) that use proteomic analysis for biomarker detection in HCC?


c). The authors have done a good job of organizing the paper in a professional manner. Publicly available raw data is clearly mentioned throughout the paper.

d) Authors should expand more on the results of figure-3 or move the figure to supplementary. I do not think this figure adds much to the study.

Experimental design

The article Fits well within the Aims and Scope of the journal. The following minor improvements should be made to the manuscript :

a) I am worried a little looking at 95% CI HC for CDC45. Can the authors try cross-validation to see if the standard error for CDC45 is stable in both univariate and multivariate cox regression?

Validity of the findings

The authors have done a good job of providing all underlying data needed to replicate the study. The tables and figures are sufficiently clear to understand the results. The following improvements can be made to the manuscript :

1) I am curious why did authors specifically study CDC45 besides it is reported in the literature since TMA can be performed on multiple genes at once. The authors should add a line expanding on this.

2) Inline 226-27, the claim that " correlation between CDC45 and HCC is related to these functions". I don't think GO and KEGG analysis warrants such speculation.

Additional comments

I think the authors have done a good job of presenting the prognostic value of CDC45 in this paper.
The author's studied the prognostic implications of CDC45 expression using IHC and TMA in HCC. Overall, results and methodology was presented in a clear and concise manner and are of interest to a broader set of audiences. I am curious why did authors specifically study CDC45 besides it is reported in the literature since TMA can be performed on multiple genes at once. (authors should add a line expanding on this)
Since the author's claim that "CDC45 expression is lower in Tumor cells" compared to previous studies, I am wondering whether this can be attributed to comparisons to adjacent normal tissues instead of healthy controls? The authors should discuss this issue/limitation in the discussion section.

While I appreciate the effort, if authors could do some below minor revisions, that would greatly improve the paper.
1) Expansion of discussion of results of figure-3 or removal if needed.
2) Check standard error stability by cross-validation for CDC45
3) Add a statement about why authors specifically chose CDC45.

---

## Round 0.2 · accepted · Accept

Please, even though the paper is accepted, provide the passage number of the cells and the density of cell seeding when you check the proofs before publication.

Reviewer 1 ·

Basic reporting

Unfortunately, this result cannot be approved unless the relationship with prognosis can be proved by WB or PCR using HCC tissue.

Experimental design

Unfortunately, this result cannot be approved unless the relationship with prognosis can be proved by WB or PCR using HCC tissue.

Validity of the findings

Unfortunately, this result cannot be approved unless the relationship with prognosis can be proved by WB or PCR using HCC tissue.

Additional comments

Unfortunately, this result cannot be approved unless the relationship with prognosis can be proved by WB or PCR using HCC tissue.

Reviewer 2 ·

Basic reporting

English writing has been improved.

Experimental design

Please provide the passage number of the cells and the density of cell seeding.

Validity of the findings

No comment

Additional comments

The authors have responded to the questions well. The manuscript has been greatly improved.

Reviewer 3 ·

Basic reporting

na

Experimental design

na

Validity of the findings

na

Additional comments

The authors have addressed my comments and made the required modifications.

·

Basic reporting

This paper is well-organized with good background material. The authors have addressed all my previous comments

Experimental design

The article Fits well within the Aims and Scope of the journal. The authors did address my comment about 95% CI and performed a lasso regression

Validity of the findings

The authors have done a good job of providing all underlying data needed to replicate the study. The tables and figures are sufficiently clear to understand the results

Additional comments

I think the authors have greatly improved the paper by addressing concerns by the reviewers